# A Single Center Retrospective Review of Patients from Central Italy Tested for Melanoma Predisposition Genes

**DOI:** 10.3390/ijms21249432

**Published:** 2020-12-11

**Authors:** Paola De Simone, Irene Bottillo, Michele Valiante, Alessandra Iorio, Carmelilia De Bernardo, Silvia Majore, Daniela D’Angelantonio, Tiziana Valentini, Isabella Sperduti, Paolo Piemonte, Laura Eibenschutz, Angela Ferrari, Anna Carbone, Pierluigi Buccini, Alessandro Paiardini, Vitaliano Silipo, Pasquale Frascione, Paola Grammatico

**Affiliations:** 1Oncologic Dermatology, San Gallicano Dermatological Institute IRCCS, 00144 Rome, Italy; alessandra.iorio@ifo.gov.it (A.I.); paolo.piemonte@ifo.gov.it (P.P.); laura.eibenschutz@ifo.gov.it (L.E.); angela.ferrari@ifo.gov.it (A.F.); anna.carbone@ifo.gov.it (A.C.); pierluigi.buccini@ifo.gov.it (P.B.); vitaliano.silipo@ifo.gov.it (V.S.); pasquale.frascione@ifo.gov.it (P.F.); 2Laboratory of Medical Genetics, Department of Molecular Medicine, Sapienza University, San Camillo-Forlanini Hospital, 00100 Rome, Italy; i.bottillo@gmail.com (I.B.); micvaliante@gmail.com (M.V.); carmeliliadebernardo@uniroma1.it (C.D.B.); silviamajore@uniroma.it (S.M.); danieladangelantonio@uniroma1.it (D.D.); tizianavanetini@uniroma1.it (T.V.); paola.grammatico@uniroma1.it (P.G.); 3Biostatistical Unit, Regina Elena National Cancer Institute IRCCS, 00100 Rome, Italy; isabella.sperduti@ifo.gov.it; 4Department of Biochemical Sciences “A. Rossi Fanelli”, Sapienza University, 00100 Rome, Italy; alessandropaiardini@uniroma1.it

**Keywords:** familial melanoma, multiple primary melanoma, melanoma susceptibility genes

## Abstract

Cutaneous malignant melanoma (CMM) is one of the most common skin cancers worldwide. CMM pathogenesis involves genetic and environmental factors. Recent studies have led to the identification of new genes involved in CMM susceptibility: beyond CDKN2A and CDK4, BAP1, POT1, and MITF were recently identified as potential high-risk melanoma susceptibility genes. This study is aimed to evaluate the genetic predisposition to CMM in patients from central Italy. From 1998 to 2017, genetic testing was performed in 888 cases with multiple primary melanoma and/or familial melanoma. Genetic analyses included the sequencing CDKN2A, CDK4, BAP1, POT1, and MITF in 202 cases, and of only CDKN2A and CDK4 codon 24 in 686 patients. By the evaluation of the personal and familial history, patients were divided in two clinical categories: “low significance” and “high significance” cases. 128 patients (72% belonging to the “high significance” category, 28% belonging to the “low significance” category) were found to carry a DNA change defined as pathogenic, likely pathogenic, variant of unknown significance (VUS)-favoring pathogenic or VUS. It is important to verify the genetic predisposition in CMM patients for an early diagnosis of further melanomas and/or other tumors associated with the characterized genotype.

## 1. Introduction

Cutaneous malignant melanoma (CMM) is one of the most common skin cancers worldwide [1]. Most of the patients experience the occurrence of a single CMM during their life (single primary melanoma, SPM); nevertheless, multiple primary melanomas (MPMs) occur in up to 8.2% of the cases both in a synchronous or metachronous manner [2]. CMM incidence has been found to vary according to the population’s origin: in Europe, northern countries show a higher CMM incidence compared to southern countries [3]. Likewise, in northern Italy, the incidence rates of melanoma is higher than the southern parts of the country [3]. CMM pathogenesis involves genetic and environmental factors [4,5]. Among the susceptibility genetic factors, germline alterations of the cyclin-dependent kinase inhibitor 2A (CDKN2A) gene are the most recognized cause of CMM hereditability within populations of different geographic areas [6]. CDKN2A is a tumor-suppressor gene that is involved in the regulation of cell cycle and that encodes four transcript variants: p16INK4A, p14ARF, p12, and p16γ [7]. p16INK4A and p16γ encode structurally related isoforms known to function as inhibitors of the CDK4 kinase, while p14ARF includes an alternate first exon located 20 Kb upstream of the remainder of the gene. This isoform functions as a stabilizer of the tumor suppressor protein p53 as it can sequester the E3 ubiquitin-protein ligase MDM2, a protein responsible for the degradation of p53 [8]. Somatic CDKN2A loss, by deletions or by promoter’s hypermethylation, has been shown to be a significant event in a number of cancer types [8].

In addition to CDKN2A, some CMM pedigrees have been found to carry mutations of the cyclin-dependent kinases-4 (CDK4) gene, at the binding site (i.e., CDK4 codon 24) for the CDKN2A gene product [9,10]. Moreover, recently, some studies have implicated other predisposing *loci* in the susceptibility to CMM [11]. These genes include BAP1 (BRCA1-associated protein 1), *MITF* (microphthalmia-associated transcription factor), and POT1 (protection of telomeres 1) [12,13,14,15,16]. Germline BAP1 mutations are indeed associated with a cancer syndrome characterized by an increased risk of malignant mesothelioma, atypical melanocytic tumors (melanocytic BAP1-mutated atypical intradermal tumors), uveal and cutaneous melanoma, and other neoplasms [13]. The *MITF* variant p.E318K is associated with both familial and sporadic melanoma susceptibility and/or renal cell carcinoma risk [12,14,15,17]. Finally, mutations in POT1 gene are associated with familial melanoma [16], glioma [18], chronic lymphocytic leukemia [19], mantle cell lymphoma [16], and cardiac angiosarcoma [20]. In 2015, a revision of the prevalence of mutations in susceptibility CMM genes reported that, in 2511 melanoma-prone pedigrees, the mutation prevalence was about 20% for CDKN2A, 0.7% for CDK4, 1% for BAP1, and 0.5% for POT1 [17]. The prevalence in Italy of the MITF p.E318K, is estimated around 2% [21]. However, it is known that, at least for CDKN2A, the mutation frequency can range from 5% to 72% depending on the patients’ ethnic background, UV exposure, and selection criteria [22,23,24,25,26,27,28,29,30]. Consequently, the use of genetic testing for melanoma predisposition genes in clinical practice has been controversial [6,31]. In 2009, Leachman et al. proposed that, in regions with a moderate to high incidence of melanoma, the higher rates of CDKN2A positivity is reached in individuals abiding the rule of 3: cases with 3 or more primary invasive melanomas and/or families with at least one invasive melanoma and two or more other diagnoses of invasive melanoma and/or pancreatic cancer among first- or second-degree relatives on the same side of the family [31]. Conversely, in countries with a low melanoma incidence, the selection criteria for genetic counseling should follow the rule of two: individuals with two primary melanomas and/or families with at least one invasive melanoma and one or more other diagnoses of melanoma and/or pancreatic cancers among first- or second-degree relatives on the same side of the family [31].

The aim of our study was to carry out an evaluation of the mutation rate of CDKN2A, CDK4, BAP1, MITF, and POT1 genes in a large cohort of CMM patients from central Italy, distinguished according the familial and clinical history.

## 2. Patients and Methods

At the Melanoma Unit of San Gallicano Institute in Rome, starting from 1998, about 10,000 patients with CMM have been evaluated. Among them, 888 (9%) cases with (MPMs) and/or familial melanoma (FM) were referred to genetic counselling. The FM criteria was assigned to cases with at least two verified cases of melanoma in first- or second-degree relatives. Those 888 selected cases included: 309 MPMs patients, 435 FM patients, and 144 cases with both MPMs and FM. 

By the evaluation of the personal and familial history, those case were divided in two clinical categories: (i) low significance cases: subjects who developed two melanomas; or patients with only one melanoma together with a first or a second-degree family member affected by the same cancer; (ii) high significance cases: subjects who developed at least three melanomas; or patients who developed two melanomas together with at least a first or a second-degree family member affected by one melanoma; or subjects diagnosed with only one melanoma together with at least two first or second-degree family members affected by the same cancer.

### 2.1. Molecular Analyses

After obtaining patients’ informed consent, blood samples were taken from all index cases, as well as from additional family members if available. DNA was extracted with standard techniques and used for genetic testing.

Between the years 1998 and 2017, 686 melanoma cases were referred to the genetic analysis of CDKN2A gene and of CDK4 codon 24 by Sanger sequencing. We named this testing approach *Sanger-Test*. The primers employed for the PCR amplification are given in Table 1. In 2018, 202 cases that were referred to genetic test and were analyzed by next generation sequencing (NGS). By this approach, the CDKN2A, CDK4, BAP1, MITF, and POT1 genes were analyzed on the Ion S5XL System (ThermoFisher, Carlsbad, CA, USA). The mapped reads were analyzed by means of the Variant Caller v5.4.0.46 plugin (ThermoFisher).

By a retrospective evaluation of all the molecular data collected between the years 1998 and 2018, the identified DNA variants were classified according to the ACMG/AMP 2015 guidelines [32] modelled by a Bayesian framework as previously described [33,34]. This approach allowed a better definition of the DNA variants into seven classes: (i) pathogenic (P), (ii) likely pathogenic (LP), (iii) Variants of Unknown Significance (VUS)-favoring pathogenic, (iv) VUS, (v) VUS-favoring benign, (vi) likely benign, and (vii) benign. All the variant classified as pathogenic, likely pathogenic (VUS-3B), VUS-favoring pathogenic, and VUS were confirmed by Sanger sequencing.

### 2.2. Molecular Modelling

Protein structures and isoforms were experimentally determined by X-ray crystallography [35], or were inferred by homology modeling. Protein models for the p16 isoforms were built using the homology modeling approach implemented in modeler-9 package [36]. PSI-BLAST was used to find suitable structural templates for each sequence to model [37]. The sequences of each protein target to model and its structural template were then aligned by using the program CLUSTALW [38]. Ten different models were built for each target protein and evaluated using several criteria. The model displaying the lowest objective function was taken as the representative model. Mutations on protein structures were carried out using the “Mutate model” script implemented in PyMod 2 [39]. The script takes as input a given three-dimensional structure of a protein and mutates a single residue. The residue sidechain position is then optimized by energy minimization and refined by molecular dynamics simulations.

## 3. Results

After clinical evaluation and genetic counselling, 538/888 (60%) cases were classified in the “low significance” class, while 350/888 (40%) fulfilled the “high significance” criteria. The cohort description of all cases referred to genetic testing is given in Table 2A. In both classes, there were no differences neither in patients’ sex distribution, nor in the age at first melanoma onset. A slightly higher percentage of “high significance case” showed the presence of other cancers in addition to melanoma. The type and distribution of these tumors is shown in Figure 1.

In total, 128/888 (14%) of cases were found to carry a DNA variant defined as Pathogenic (P), Likely Pathogenic (LP), VUS-favoring pathogenic (VUS-3B), or VUS. In particular, 98 cases carried a variant in CDKN2A, 10 in MITF, 9 in CDK4, 9 in BAP1 in 2 in POT1. Most of the patients (i.e., 92/128, 72%) carrying those genetic changes, belonged to the “high significance” category (Table 2B). The age at first melanoma onset and the number of cases with other cancer in addition to CM (Table 2B) were not significantly different compared to the data of the entire cohort referred to genetic testing (Table 2A).

Among the 538 “low significance” patients, 418 underwent the Sanger-Test and 13 (3%) were found to carry a P, LP, VUS-3B, or VUS DNA variant. Conversely, between the 120 cases analyzed by the NGS-Test, positive cases were 23 (19%).

Regarding the 350 “high significance” patients, 268 underwent the Sanger-Test and 82 the NGS-Test. Sixty-seven (25%) cases were positive to the Sanger-Test, while 25 (30%) were positive to the NGS-Test. The spectrum of genetic alterations found in both classes of patients by the use of the two testing approaches is shown in Figure 2.

Most of the cases that resulted positive to genetic testing were “high significance” patients with alterations in CDKN2A. Pathogenic and other variants with different degrees of pathogenic evidences (i.e., LP and VUS-3B) were identified only in CDKN2A, *CDK4*, and MITF genes (Figure 3).

In total, 33 different P, LP, VUS-3B, or VUS DNA changes were identified in CDKN2A gene, with the p16 c.71G > T (p.R24P), the c.212A > G (p.N71S), and the c.301G > T (p.G101W) recurring each one in more than 10 patients. By molecular modelling, each of the CDKN2A missense variants resulted in the structural alteration of at least one CDKN2A isoform (Table 3). Each of the identified variants was mapped over the CDKN2A gene structure and over its 4 isoforms (i.e., p16, p14, p16γ and p12) (Figure 4). Figure 5 shows the molecular modelling of missense CDKN2A changes. The list of the identified *CDKN2A* variants and the results of the molecular modelling over the 4 CDKN2A isoforms are given in Table 3.

The variants identified in MITF gene included the pathogenic c.952G > A (p.E318K) already reported as a risk factor for CM development and 3 VUS.

Regarding the three variants identified in CDK4, two were pathogenic and mapped in exon 2 codon 24 (i.e., the c.71G > A (p.R24H) and the c.71G > T (p.R24L)). The third variant was an in frame deletion mapping in exon2 codon 48 (i.e., the c.132_134delAGG (p.G48del)) and classified as VUS.

## 4. Discussion

Cutaneous melanoma is a potentially very lethal neoplasm if not diagnosed and treated early. Over the past few decades, there has been a significant increase in the incidence of this cancer, regardless of age. In light of these considerations, it is essential to identify the subjects of the general population at high risk of cutaneous melanoma. The main factors to consider to establish an individual’s overall risk for the development of cutaneous melanoma are: skin risk factors (phototype, nevi number, atypical nevi number), environmental/behavioral factors, genetic predisposition.

Numerous genes have been associated with melanoma: beyond CDKN2A and CDK4, primarily included as high-risk genes for familial melanoma, BAP1, POT1, and MITF were recently also identified as potential high-risk melanoma susceptibility genes.

Therefore, this condition inevitably predisposes a multidisciplinary approach for the appropriate and correct management of familial and multiple melanomas, based on the formulation and processing of a specific plan for patients affected by MPM or FM (Figure 6).

In this regard, it is necessary to check the patient periodically not only to identify recurrence or metastasis but also to detect the appearance of new primary melanomas.

Dermatological examination is crucial and represents the primary step in the management program.

Given that new primary melanoma can occur in a skin site, it is important to examine the entire skin surface during each follow-up, with a clinical full body skin examination which includes the scalp, oral mucosa, genital area, and nails. 

The nevi with changes in morphology, size, and dermoscopic atypia and new atypical lesions must undergo surgical excision and histological examination. 

The appropriate follow-up frequency for risk subjects is based on the clinical and dermoscopic risk factors identified from dermatological examination.

In this study, the assessment of hereditary predisposition to CMM was carried out in the context of genetic counselling, to which all patients with FM and MPM were candidate. Genetic counselling was then performed and consisted of two stages:Pre-test counselling: collection of familial and personal anamnestic data and histological data, for formulating an in-depth ≥3-generation profile and for assessing cases that were candidates to genetic testing. This step also included an explanation of the objectives, possible implications, and test limitations.Post-test counselling: interpretation of the results of the genetic test.

The evolution of DNA sequencing technologies (i.e., NGS-Test) allowed the inclusion of additional genes in the melanoma genetic test. Since the NGS-Test has been introduced in recent times in the here presented diagnostic flow, only about 23% of the cases were analyzed by this more sensitive approach. However, the “preliminary” data about the Sanger/NGS switch, showed that the evolution of the testing approach let a decrease of the proportion of negative results, even if CDKN2A is confirmed to be the major CMM gene (Figure 2). At CDKN2A, the data on LP variants is much stronger than P and VUS ones (Figure 3), inferring that, at present, most of the DNA changes identified in CDKN2A do not meet enough criteria to be definitely considered pathogenic. This observation may reflect the lack of penetrance of CDKN2A mutations (e.g., a DNA variant that segregates in unaffected individuals at the time of clinical evaluation) and/or the lack of large multigenerational genealogies to be studied. The variants identified in CDKN2A map along the entire coding sequence of the gene, with the c.71G > T (p.R24P), the c.212A > G (p.N71S) and the c.301G > T (p.G101W) that could stand as mutation hot-spots in the Italian population (Figure 4). Indeed, the c.301G > T is reported in ClinVar database as the most common founder pathogenic CDKN2A variant seen in numerous geographically different families (ClinVar Accession: VCV000009412.11). Most of the variants involve more than one CDKN2A isoform (Figure 4) and the molecular modeling predictions demonstrated that 17 among 20 missense changes may destabilize the protein structure of at least one isoform (Table 3 and Figure 5). Considering that the different CDKN2A isoform have different tissues expression, we tested the molecular and clinical data for genotype-phenotype correlations/associations. However, no statistically significant result was obtained (data not shown).

At least 5 enhancers are reported in the CDKN2A promoter region [8], but they were not evaluated in the present study. Therefore, in patients negative for mutations in the coding sequence of CDKN2A, we cannot exclude the presence of genetic/epigenetic alterations in those regulatory regions.

Though most of the CDKN2*A* alterations were identified in “high significance” cases by both the Sanger-Test and the NGS-Test, the inclusion of BAP1, MITF, and POT1 in the genetic testing approach, allowed a greater reduction of negative cases in the “low significance” cohort than in the “high significance” population (Figure 2). As showed in Figure 3, pathogenic or favoring-pathogenic variants were found only in CDKN2A, CDK4, and MITF, with CDK4 mutations present just in “high significance” cases and the MITF p.E318K found in both “high” and “low” significance cases. Conversely, in BAP1 and POT1 only VUS variants were detected. This finding suggests that, at least in cases from central Italy, larger studies are need to assess the involvement of BAP1 and POT*1* in the susceptibility to CMM and their possible association to additional cancers. Anyhow, the investigation of the MITF p.E318K should be performed also to subjects who developed only two melanomas, or even a single melanoma together with a first or a second-degree family member affected by the same cancer. Since genetic alterations of BAP1, MITF, and POT1 are associated also to other cancer types [12,13,14,15,16,17,19,20], we propose for these patients an instrumental follow-up aimed at the possible early recognition of other associated neoplasm (Figure 6):

(i) In the case of genetic testing with a negative result, the patient undergoes clinical and instrumental follow-ups every 8–10 months. Molecular analysis of further genes related to increased neoplastic susceptibility should be performed if the family history is indicative of another possible genetic syndrome. Otherwise, we recommend new genetic counselling after about 12 months to proceed with an overall reassessment of the personal and family history, eventually leading to new genetic tests related to CM and/or to reassess the possible VUS-3A variant.

(ii) The patient whose genetic testing results in the identification of a VUS undergoes high-level skin surveillance inclusive of clinical and instrumental revaluation every 6 months. If possible and appropriate, the variant’s family segregation study is carried out for a research purpose. The interpretation of the VUS is also checked over time (at least every 6 months).

(iii) In the case of a positive result (i.e., identification of P or LP variant or of a VUS-3B), specific surveillance is scheduled as shown in (Figure 6).

CDKN2A: The patient undergoes intensive-level skin surveillance inclusive of clinical and instrumental follow-ups every 3 months; in the presence of familial pancreatic cancer (I-II degree) the patient is referred for gastroenterological evaluation and magnetic resonance imaging (MRI) of pancreas and/or an endoscopic ultrasound (ES). Familial surveillance linked to genetic counselling is also recommended. The absence of familial pancreatic cancer does not require a gastroenterological and instrumental screening test, even if some patients do it anyway.

CDK4: Given the probability of developing multiple melanoma equivalent to about 40% [40], we suggest to carry on clinical and instrumental skin re-evaluation with a 3-month follow-up frequency.

BAP1: We propose to proceed with biannual check-ups starting from the age of 18 years old. Check-ups should include clinical and instrumental skin examination. With regard to uveal melanoma (UM), recommendations indicate an annual eye examination starting at 16 years of age (younger age considered if there is a family member with early UM) which consists in direct and indirect ophthalmoscopy, fundus photography, and ocular ultrasound [41]. For malignant mesothelioma and renal cell carcinoma surveillance, we suggest an annual physical examinations and respiratory evaluation and eventual instrumental ES investigations (renal, abdomen, and chest) or MRI (abdomen and chest) are performed every 2 years between the age of 30 and 55 years; after 55 years of age, it is advisable to carry out a biennial abdomen and chest computed tomography (CT) or an abdomen and chest MRI with contrast and a renal and chest ES in between [41].

MITF: The patients undergoes clinical and instrumental skin revaluation with a 6-month follow-up frequency abdomen ultrasound evaluation. The patients also undergo urinary cytology every 6 months [15]. In the presented cohort, one patient belonging to the “low significance” category and carrying the *MITF* p.E318K, developed renal cancer in addition to CMM and also had a positive family history for that disease.

POT1: Follow-up is comparable to that provided for Li-Fraumeni syndrome [42,43].

In general, we also propose that individuals carrying CDK4, BAP1, MITF, or POT1 mutations and showing a high number of nevi, should undergo a higher frequency of checks, regardless of the type of mutated gene. As already known, in the case of positive genetic responses, it is necessary to carry out genetic testing in the proband’s family members, regardless of the mutated gene. On the other hand, in the case of negative genetic responses, we propose that the proband’s first degree kindred should undergo clinical and instrumental skin evaluation with an annual follow-up frequency.

## 5. Conclusions

Recent advances in technology have led to the identification of new genes involved in melanoma susceptibility. This has made it possible to explain less than 30% of the genetic susceptibility in melanoma-prone families. It is very important that new predictive biomarkers and prognostic factors in CM are identified. This can be achieved through the correlation between genotype, phenotype (fair skin type, solar freckles, and multiple and atypical nevi), and environmental risk factors (intense and intermittent sun exposure). It is also crucial that all patients with familial and multiple melanomas receive genetic counselling for the identification of high-risk subjects requiring targeted follow-up.

The ultimate objective is to be able to create a personalized follow-up program that takes into account the patient’s clinical and biological risk factors and exposure to environmental factors. To achieve this target, it is essential to develop a multidisciplinary approach enabling early diagnosis and improvement in prognosis with a consequent reduction in health expenditure. The instrumental follow-up must be personalized and aimed at the possible early recognition of other associated neoplasms.

## Figures and Tables

**Figure 1 ijms-21-09432-f001:**
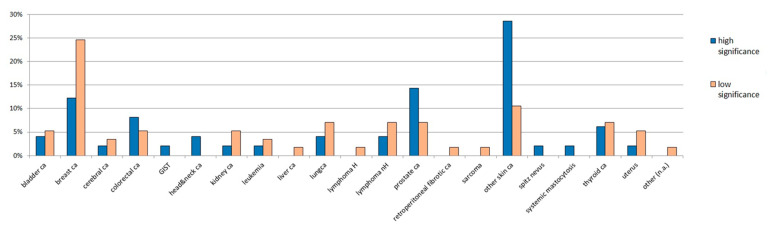
Distribution of other cancers than cutaneous melanoma in the patients’ cohort. The type of tumor is given on the *x*-axis, the percentage of patients is given on the *y*-axis. ca: cancer; H: Hodgkin; nH: non Hodgkin; n.a.: not assessed.

**Figure 2 ijms-21-09432-f002:**
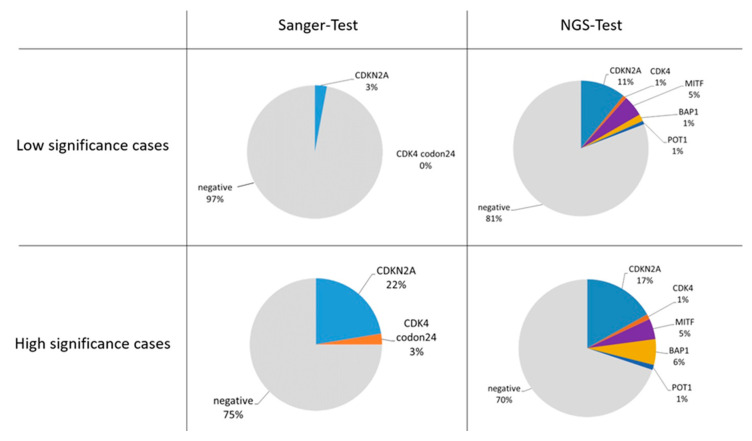
Spectrum of genetic alterations identified in “low significance” and “high significance” cutaneous melanoma patients by the use of both Sanger and NGS testing approach. The percentage of cases is given nearby each data series represented in the pie charts.

**Figure 3 ijms-21-09432-f003:**
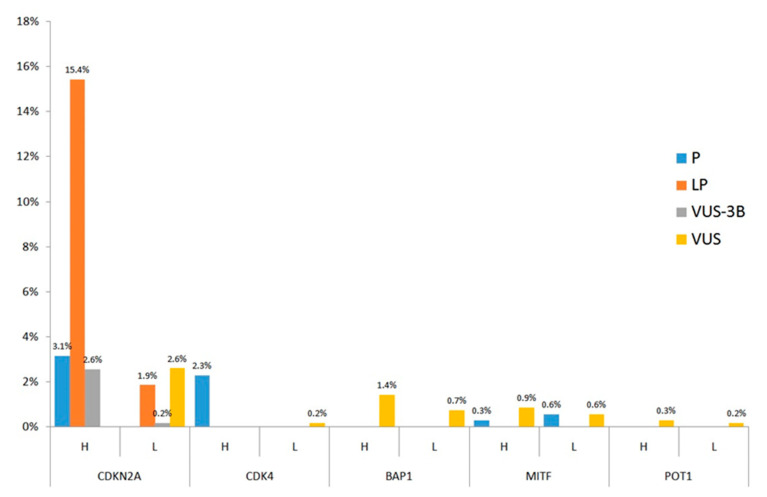
Patients with positive genetic testing results. The proportion of “high significance” and “low significance” patients carrying Pathogenic (P), Likely Pathogenic (LP), VUS-favoring pathogenic (VUS-3B), and VUS variants is shown for each of the analyzed genes. H: high significance patients, L: low significance patients.

**Figure 4 ijms-21-09432-f004:**
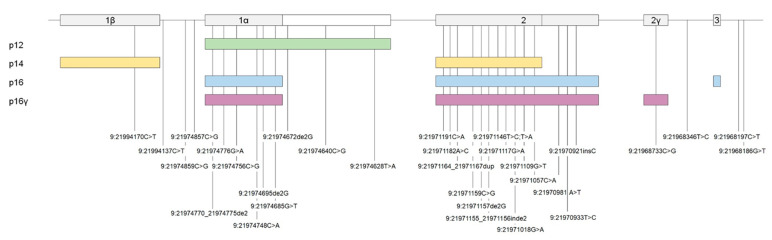
CDKN2A variants mapped over the genic structure and over the 4 coded isoforms. Exons are in scale and are represented as grey rectangles on the top. Introns are not in scale and they are represented by a black line through the genic sequence. The CDKN2A isoforms (i.e., p12, p14, p16 and p16γ) are shown on the bottom as colored rectangles. The DNA variants are name according the hg19 Human Genome Assembly.

**Figure 5 ijms-21-09432-f005:**
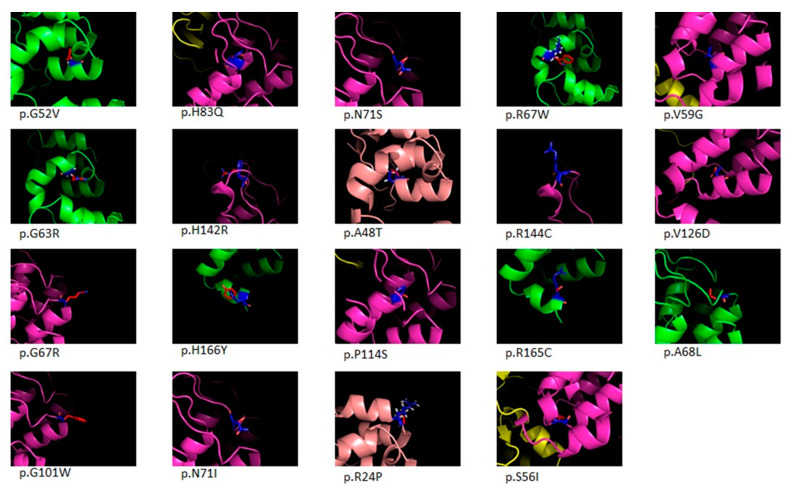
Molecular modeling analysis and localization of missense changes affecting p16 CDKN2A isoform. Proteins are represented as ribbons and colored according to protein chain. Wild-type and mutant residues are represented in blue and red sticks, respectively.

**Figure 6 ijms-21-09432-f006:**
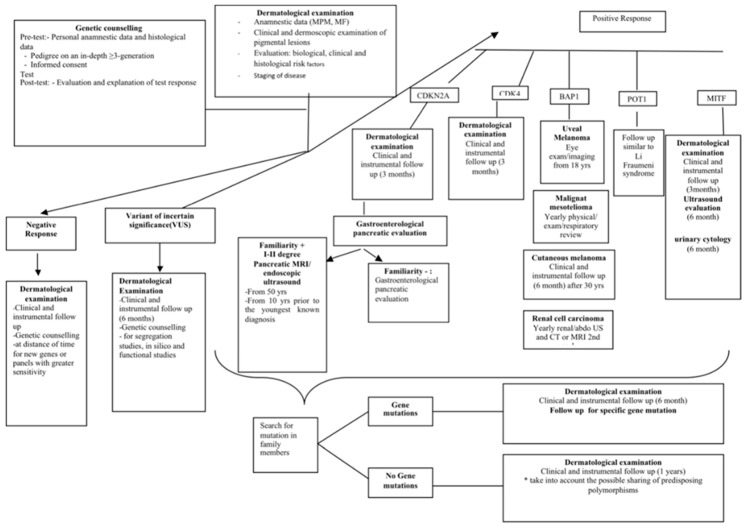
Flow chart on the therapeutic diagnostic path of the patient with familial melanoma or/and multiple primary melanoma.

**Table 1 ijms-21-09432-t001:** Primer sequence for PCR amplification and Sanger sequencing of CDKN2A gene and CDK4 codon 24 (Sanger-Test).

Amplicon	Primer Forward (5′–3′)	Primer Reverse (5′–3′)
CDKN2A exon 1α	CACCAGAGGGTGGGGCGGA	CAGGGCGTCGCCAGGAGGA
CDKN2A exon 1β	TCCCAGTCTGCAGTTAAGG	CGGGTTTACAACGACTTAGAC
CDKN2A exon 2	GGCGGTGAGGGGGCTCTACA	ACCGATTGGCGCGTGAGCTG
CDKN2A exon 3	GCCGGTAGGGACGGCAAGAG	AAAGCGGGGTGGGTTGTGGC
CDK4 codon24 (exon2)	GGATGCTGGTGGTGTTCTTT	TTATTTCCTCAGGGTCCCCA

**Table 2 ijms-21-09432-t002:** Cohort description.

**A. Cases Referred to Genetic Testing**		
	**Clinical Category**	
	**Low Significance**	**High Significance**
**Patients**	538	350
Males	254	180
Females	284	170
**Age at first melanoma onset**	47 y ± 14 y	47 y ± 15 y
Males	51 y ± 14 y	52 y ± 15 y
Females	44 y ± 13 y	42 y ± 13 y
**Patients with CM and other cancer**	50 (9%)	48 (14%)
Males	23	29
Females	27	19
**B. Cases carrying DNA variants classified as P, LP, VUS-3B, or VUS**		
	**Clinical Category**	
	**Low Significance**	**High Significance**
**Patients**	36	92
Males	11	42
Females	25	50
**Age at first melanoma onset**	44 y ± 13 y	41 y ± 14 y
Males	45 y ± 14 y	41 y ± 15 y
Females	44 y ± 13 y	41 y ± 14 y
**Patients with CM and other cancer**	2 (5%)	9 (10%)
Males	0	5
Females	2	4

y: years.

**Table 3 ijms-21-09432-t003:** DNA variant identified in CDKN2A gene.

					Molecular Modelling of Missense Variants			
CDKN2A Variant	Genomic Position (hg19)	Coding Impact	No. of Carrier Patients	ACMG/AMP Classification	p16	p16γ	p12	p14
p14:c.161G > A (p.R54H)	chr9:21994170	missense	2	VUS-3B	-	-	-	n.a.
p14:c.193 + 1G > A	chr9:21994136	Splicing	7	P				
p16:c.-31G > C	chr9:21974857	5′UTR	1	VUS				
p16:c.-19G > C	chr9:21974845	5′UTR	1	VUS				
p16,p12,p16γ:c.51C > T (p.A17A)	chr9:21974776	silent	1	VUS				
p16,p12,p16γ:c.52_57delACGGCC (p.T18_A19del)	chr9:21974770_21974775	in frame deletion	2	LP				
p16,p12,p16γ:c.71G > C (p.R24P)	chr9:21974756	missense	17	LP	Hd	Hd	Hd	-
p16,p12,p16γ:c.79G > T (p.E27X)	chr9:21974748	truncating	1	P				
p16,p12,p16γ:c.132delC (p.Y44X)	chr9:21974695	truncating	1	P				
p16,p12,p16γ:c.142C > A (p.P48T)	chr9:21974685	missense	3	LP	Hd	Hd	Hd	-
p16,p16γ:c.150 + 5G > T; p12:c.155G > T (p.G52V)	chr9:21974672	splicing/missense	1	VUS	-	-	Hd	-
p16,p16γ:c.150 + 37G > C; p12:c.187G > C (p.G63R)	chr9:21974640	splicing/missense	3	VUS	-	-	Hd	-
p16,p16γ:c.150 + 49A > T; p12:c.199A > T (p.R67W)	chr9:21974628	splicing/missense	1	VUS	-	-	Sd	-
p16,p16γ:c.167G > T (p.S56I); p14:c.210G > T(p.Q70H)	chr9:21971191	missense	3	LP	D	Ss	-	n.a.
p16,p16γ:c.176T > G (p.V59G); p14:c.219T > G(p.S73R)	chr9:21971182	missense	4	LP	Hd	Hd	-	n.a.
p16,p16γ:c.191_194dupTGCT (p.H66Afs*55);p14: c.234_237dupTGCT (p.P80Cfs*82)	chr9:21971164_21971167	truncating	1	P				
p16,p16γ:c.199G > C (p.G67R);p14:c.242G > C(p.R81P)	chr9:21971159	missense	3	VUS-3B	Hd	Hd	-	n.a.
p16,p16γ:c.201delC (p.A68Rfs*78); p14:c.244delC (p.R82Afs*90)	chr9:21971157	truncating	1	P				
p16,p16γ:c.202_203GC > TT (p.A68L); p14: c.245_246delGCinsCT (p.R82P)	chr9:21971154	missense	2	VUS-3B	Hd	Hd	-	n.a.
p16,p16γ:c.212A > G (p.N71S); p14:c.255A > G (p.Q85Q)	chr9:21971146	missense	11	LP	Sd	Hd	-	n.a.
p16,p16γ:c.212A > T (p.N71I); p14:c.255A > G (p.Q85H)	chr9:21971146	missense	5	LP	Ss	Ss	-	n.a.
p16,p16γ:c.249C > A (p.H83Q); p14:c.292C > A (p.R98R)	chr9:21971109	missense	2	VUS-3B	D	Hd	-	n.a.
p16,p16γ:c.301G > T (p.G101W); p14:c.344G > T (p.R115L)	chr9:21971057	missense	16	LP	Hd	Hd	-	n.a.
p16,p16γ:c.340C > T (p.P114S); p14: c.383C > T (p.A128V)	chr9:21971018	missense	1	LP	Hd	Hd	-	-
p16,p16γ:c.377T > A (p.V126D)	chr9:21970981	missense	1	LP	Hd	Hd	-	-
p16,p16γ:c.425A > G (p.H142R)	chr9:21970933	missense	1	VUS-3B	Sd	N	-	-
p16,p16γ:c.430C > T (p.R144C)	chr9:21970928	missense	1	VUS	Sd	N	-	-
p16,p16γ:c.436_437insG (p.D146Gfs*19)	chr9:21970922	truncating	1	LP				
p16γ:c.496C > T (p.H166Y)	chr9:21968732	missense	1	VUS	-	Ss	-	-
p16:c.458-105A > G (p.156_157del)	chr9:21968346	in frame deletion	2	VUS-3B				
p16γ:c.495G > C (p.R165S)	chr9:21968733	missense	1	VUS	-	D	-	-
p16:c.*31G > A	chr9:21968197	3′UTR	1	VUS				
p16:c.*42C > A	chr9:21968186	3′UTR	1	VUS				

No.: number; Hd: highly destabilizing; D: destabilizing; Sd: slightly destabilizing; Ss: slightly stabilizing; N: neutral; n.a.: not assessed.

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
