# Peer review of "A Single Center Retrospective Review of Patients from Central Italy Tested for Melanoma Predisposition Genes"

_ijms, 2020, doi:10.3390/ijms21249432_

Round 1

Reviewer 1 Report

This study aims to identify genetic markers for the prediction and prognosis of melanoma using retrospective data from one central Italy research center. The author grouped all patients into two groups, high and low risk, based on family history, and tried to associate the risk groups with the variants and mutations of target DNA.

The goal of this study is good, but the finding is limited. Only 128 patients out of the 888 patients carry one type of DNA change, even 72% are in the high-risk group. There is no definite recommendation that could draw from this study. An increased sample size may not provide more information. However, even negative data deserve in-depth discussion. The authors present all the figures with data but a minimal discussion. For each figure, conclusions and discussion help readers understand the study design beyond the legend only.

Figure 1, what is the conclusion to present the distribution of other cancers than cutaneous melanoma in the patients' cohort.  

Figure 2, most of the data from 200 patients with a more sensitive method. Is it suitable to compare these two sets of data together? At least a discussion of the potential problem is warrantied.

Line134-137, the finding, and the suggestion do not seem logical. May the author provide more background information to justify further large study is needed for BAP1 and POT1? Did any other types of cancer have a trend on these two genes?

Figure 3, the data on LP is much stronger than P and VUS at CDKN2A. A discussion will be excellent for the reader. Does the grouping, p, LP, VUS-3B, and VUS, tell a story here?  

Figure 4,  are any variants mapped associated with different disease outcome levels? Or they all carry the same weight in counting for FM?

Figure 5, the authors could point out the conclusion from all the beautiful pictures in addition to the blue and red sticks?  

In figure 6, many decisions may need a discussion or explanation to let the reader know the study design. For example,  in Line 165-167, Is there any real benefit with further testing from this study? Line 177-178, what is the rationale of these further testing? May some references provide clear support for the decision or conclusion? The justification for CD4 is reasonable. The same reasoning is needed for Line190-192.

Line 193-197, the MITF, one patient out of all the cases, developed renal cancer. Is this by chance or supported by any scientific reference?

From Line 209, the conclusion, I saw genotype throughout the paper, but I don't see phenotype, nor environmental risk factors presented.

Line 211-212, high-risk subjects is the goal of this study using known genes. But with so many subtypes and other cancers, I am not sure the grouping tells the story of disease prevention or diagnostic function.

For Line 212, the "personalized follow-up program" is a good goal but not related to the data presented here

Author Response

We do not find a match between the lines of the text and your comments. We would like to resubmit the original text as there was probably a
submission error with loss of part of the text.
we are sorry to cause annoyance. we look forward to your directions
Best regards

Reviewer 2 Report

The article is well written and of high interest!

However, it needs some additions. At least in introduction, the authors should re-evaluate their statements from lines 43-72 based on analysis of data from GeneCard (https://www.genecards.org/cgi-bin/carddisp.pl?gene=CDKN2A). A discussion about enhancers of CDKN2A will be highly appreciated. Don’t forget to correctly cite GeneCard, according to their policies: https://www.genecards.org/Guide/Publications

Fig. 6: make another print screen with Spelling and Grammar disabled (un-select the “Check grammar” option)

Author Response

We do not find a match between the lines of the text and your comments. We would like to resubmit the original text as there was probably a
submission error with loss of part of the text.
We are sorry to cause annoyance. We look forward to your directions
Best regard  

Reviewer 3 Report

the manuscript reports the results of cases of cutaneous melanoma classified by clinical risk categories (low and high significance) and genetic  alterations.

the manuscript is interesting, clear and well written, but the overall presentation needs to be improved 

- in results section it is not reported in how many cases was found a genetic alteration (of the different genes evaluated);

actually, in figure 2 and figure 3 are reported the percentage values, the large table 2 reports the cases of variants in CDKN2A gene,

I suggest to add some simple sentences about the number of patients with alterations in each gene; this could help the reader to evaluate the results (in general the number of patients with alterations of CDKN2A, CDK4, BAOP1, POT1, ... should be reported)

- the figures and tables should be placed into the text at the appropriate position, to help the reader to understand the results, and not at the end of the article. (but figure 3 reporting particular results for a speficic gene can be left at the end of the article)

in particular table 2, which reports the cohort data, should be placed at the beginning of the results section of the manuscript

in the text should be added the indications for figure and tables at the appropriate position, connected with a pertinent text: it is reported only for table 1 (line 95), fig 2 (line 130), table 6 (lines 142 and 173)

-figure 6 needs a legend

-figure 6, workflow for clinical management: it is not clear how, from the results of this study, can be suggested the time line of all controls and clinical investigations: if all these indications refer to accepted guidelines, appropriate references should be added in the text, and the text modified accordingly

Author Response

We do not find a match between the lines of the text and your comments. We would like to resubmit the original text as there was probably
a submission error with loss of part of the text.
We are sorry to cause annoyance. We look forward to your directions

Best regards